# Cistus, Acacia, and Lemon verbena Valorization through Response Surface Methodology: Optimization Studies and Potential Application in the Pharmaceutical and Nutraceutical Industries

**DOI:** 10.3390/ph17050593

**Published:** 2024-05-07

**Authors:** Filipa A. Fernandes, Márcio Carocho, Tiane C. Finimundy, Miguel A. Prieto, Isabel C. F. R. Ferreira, Lillian Barros, Sandrina A. Heleno

**Affiliations:** 1Centro de Investigação de Montanha (CIMO), Instituto Politécnico de Bragança, Campus de Santa Apolónia, 5300-253 Bragança, Portugal; filipa.alexandra.piresfernandes@uvigo.es (F.A.F.); tiane@ipb.pt (T.C.F.); iferreira@ipb.pt (I.C.F.R.F.); lillian@ipb.pt (L.B.); sheleno@ipb.pt (S.A.H.); 2Laboratório Associado para a Sustentabilidade e Tecnologia em Regiões de Montanha (SusTEC), Instituto Politécnico de Bragança, Campus de Santa Apolónia, 5300-253 Bragança, Portugal; 3Grupo de Nutrición y Bromatología, Departamento de Química Analítica y Alimentaria, Facultad de Ciencias de Ourense, Universidad de Vigo-Ourense Campus, E-32004 Ourense, Spain; mprieto@uvigo.es

**Keywords:** *Cistus ladanifer* L., *Acacia dealbata* L., *Aloysia citrodora* Paláu, extraction optimization, response surface methodology, phenolic compounds

## Abstract

*Cistus ladanifer* L., *Acacia dealbata* L., and *Aloysia citrodora* Paláu were subject to an optimization procedure for two extraction techniques (heat-assisted extraction (HAE) and ultrasound-assisted extraction (UAE)). The extracts were then analyzed by HPLC-DAD-ESI/MS for their phenolic profile (cistus—15 compounds, acacia—21 compounds, and lemon verbena—9 compounds). The response surface methodology was applied, considering four varying factors: ethanol percentage; extraction time; temperature/power; and S/L ratio, generating two responses (the major phenolic compound, or family of compounds, and the extraction yield). For cistus, both techniques optimized the extraction yield of punicalagins, with UAE proving to be the most efficient extraction method (3.22% ethanol, 22 min, 171 W, and 35 g/L). For acacia, HAE maximized the extraction of procyanidin (74% ethanol, 86 min, 24 °C, and 50 g/L), and UAE maximized the content of myricetin (65% ethanol, 8 min, 50 W, and 50 g/L). For lemon verbena, HAE favored the extraction of martynoside (13% ethanol, 96 min, 49 °C and 17 g/L) and forsythiaside UAE (94% ethanol, 25 min, 399 W, and 29 g/L). The optimal conditions for the extraction of compounds with high added value and potential for use in pharmaceuticals and nutraceuticals were defined.

## 1. Introduction

The use of plants by humans for various purposes has been rooted in humanity since ancient times. Originally, they were used not only as food but also in traditional medicine for their supposed health-promoting compounds, shelter, weapons, and many other uses [1]. Over the years, science has shown that these natural matrices produce a large number of secondary metabolites with bioactive potential, increasing the interest of industries in the valorization of natural species for food, cosmetic, pharmaceutical, and nutraceutical applications [1,2].

There are an infinite number of wild species of no apparent value that grow spontaneously in ecosystems; their exploration becomes imperative to contribute to environmental sustainability through the extraction of high-value ingredients [3]. Among the wild plants found in Europe, the Magnoliopsida (dicotyledon) plant class is widespread in Portuguese territory. There are several families in this plant class, including *Cistaceae*, *Fabaceae*, and *Verbaceae*, and although most plant species from these families are undervalued, they are associated with various health benefits [4,5].

*Cistus ladanifer* L. belongs to the *Cistaceae* family and is commonly known as cistus, being a resinous and fragrant wild plant that can grow up to 2 m tall and is widely distributed in the Mediterranean region (e.g., Portugal) [6,7]. Although it has been used since ancient times in traditional medicine (e.g., vasodilator, anti-inflammatory), perfumery (essential oil extraction), and ornamental purposes, its economic value currently refers mainly to the extraction of essential oil and labdanum resin [8].

*Acacia dealbata* L., commonly known as mimosa, belongs to the *Leguminosae* family and is considered an invasive plant that is widespread throughout the world, being dispersed throughout the Portuguese territory [9]. Although it is a highly invasive species, its economic value is still poorly explored. Currently, its exploitation is concentrated in the cellulose industry, but more and more studies show the economic potential of this species [10]. Due to its composition of secondary metabolites with bioactive potential (e.g., terpenes, phenolic compounds), the acacia species is used in traditional medicine to treat a variety of diseases, including eczema, kidney problems, fever, diabetes, pneumonia, constipation, rheumatism [2].

*Aloysia citrodora* Paláu of the *Verbenaceae* family is commonly known as lemon verbena and is geographically widespread, with a focus on Southern Europe, North Africa, and South America [11]. There are several scientific studies that demonstrate the presence of potentially health-promoting compounds (e.g., flavonoids and terpenes), reporting their use as diuretics, sedatives, antispasmodics, and digestive agents, as well as a product to prevent and/or treat insomnia, skin disorders, fever, flu, nervous problems, bronchitis, acne, etc. [11,12].

With the aim of exploiting natural resources through the valorization of species, science has invested in the use of more environmentally friendly techniques with green solvents that allow for high profitability and quality of extraction at the lowest possible cost. Thus, the new technologies (e.g., extraction assisted by ultrasound and microwaves) have been used to the detriment of conventional technologies (e.g., extraction using heat—maceration, Soxhlet), as they are sustainable methods that can be easily implemented by the industry [13,14].

The use of natural compounds is being intensively researched by science and industry to develop sustainable natural ingredients with pharmacological potential that do not incur high costs. Therefore, the application of mathematical studies to optimize the extraction of natural compounds is crucial as it reveals the optimal values of the different variables that lead to maximum extraction yield. It is important for the industry to extract compounds using environmentally friendly solvents, low time and temperature/power extraction, and S/L ratio, which contributes to the sustainability of the process.

The main objective of this work is to obtain extracts rich in phenolic compounds that can be used in various industries (pharmaceutical and nutraceuticals). Specifically, based on the literature, different experimental designs were planned, one for each of the studied matrices (cistus, acacia, and lemon verbena), using the response surface method (RSM), with the validation of the predictive models, and using two different extraction techniques (conventional—maceration; emerging—assisted by ultrasound), with different solid–liquid ratio, extraction time, temperature/power, and ethanol percentage.

## 2. Results and Discussion

### 2.1. Description of the Phenolic Profile of the Different Extracts under Study

The extracts resulting from the different extraction optimization runs were subjected to analysis by HPLC-DAD-ESI/MS. Table 1, Table 2 and Table 3 contain information regarding UV-Vis at maximum absorption, deprotonated ion, mass fragmentation, as well as the respective attempts to identify the compounds. The phenolic compounds obtained in the different extractions from the different matrices were tentatively identified by comparison with the literature and available standards. 

For cistus, a total of 15 phenolic compounds were tentatively identified through various extractions, encompassing eight ellagitannins and seven *O*-linked glycosylated flavonoids and their isomers, as detailed in the provided Table 1.

The ellagitannins in cistus are represented by peaks 1–3, 6, 7, and 13–15. Notably, peaks 1, 7, and 13 exhibited pseudomolecular ions [M-H]^−^ at *m*/*z* 1083, identified as punicalagin isomers I, II, and III. These compounds displayed characteristic fragmentation patterns with ions at *m*/*z* 781, 601, and 301, as previously described by Fernández-Arroyo et al. and Seeram et al. [15,16]. These ions likely correspond to core fragments of the compound and play a pivotal role in structural confirmation. Peaks 2, 3, 6, 14, and 15 were tentatively identified as punicalagin gallates ([M-H]^−^ at *m*/*z* 1251). These compounds exhibited analogous fragmentations with ions at *m*/*z* 1083, 781, 601, and 301, as reported in the cistus species by Saracini et al. [17]. This suggests the presence of both gallic groups and punicalagin in their structure, with the differences among isomers possibly associated with variations in the positioning of these groups. In this context, it becomes evident that gallic acid is not conjugated with punicalagin through the carboxyl group, as indicated by the *m*/*z* 1083 fragment, which corresponds to the independent loss of gallic acid itself.

Regarding flavonoids, they are represented by peaks 4, 5, and 8–12. Specifically, peaks 4, 5, 8, 10, and 12 were identified as quercetin derivatives based on their characteristic absorbance maxima around 350–360 nm, and MS^2^ fragment at *m*/*z* 301. Peak 4 ([M-H]^−^ at *m*/*z* 757) released several fragments corresponding to the losses of two hexosyl moieties (−162–162 *u*) and a pentosyl residue (−132 *u*), being attributed to a quercetin-*O*-di-hexose-pentose [18,19]. Peak 5 ([M-H]^−^ at *m*/*z* 625), with two fragments at *m*/*z* 463 and 301, indicated the loss of two hexosyl residues, respectively, and, thus, were identified as a quercetin-*O*-dihexoside with sugars positioned at different locations on quercetin. Peaks 8 ([M-H]^−^ at *m*/*z* 433) and 10 ([M-H]^−^ at *m*/*z* 447) were associated with quercetin and distinct sugars, including pentosyl (132 *u*) and desoxyhexosyl (146 *u*), respectively. Peak 12 ([M-H]^−^ at *m*/*z* 667) was identified as a quercetin-*O*-acetyl-di-hexoside, supported by the major MS^2^ fragment at *m*/*z* 301 ([M-H−324-42]^−^), representing the loss of two hexosyl residues and an acetyl residue. Peaks 9 and 11 corresponded to kaempferol derivatives, with peak 9 ([M-H]^−^ at *m*/*z* 447) being characterized as kaempferol-3-*O*-glucoside and peak 11 ([M-H]^−^ at *m*/*z* 593) as kaempferol-3-*O*-rutinoside in comparison with the commercial standards.

Barros et al. tentatively identified 27 compounds; this difference in extracted compounds is probably due to the different extraction conditions that were applied in both studies [18]. The phenolic compounds identified in cistus are widely studied. Li et al., for instance, demonstrated the potential of punicalagins as an anti-influenza agent, with the ability to inhibit neuraminidase activity, proving that these compounds are potent antivirals with a broad spectrum against influenza A and influenza B viruses [20], while Bouyahya et al. showed its antibacterial activity, with extracts extracted from cistus also correlating its composition with antidiabetic activity [21].

In acacia leaves, 21 phenolic compounds were tentatively identified, comprising 2 flavan-3-ols, 1 phenolic acid, 3 condensed tannins, 2 flavanones, 11 flavanols, and 2 flavones [22]. Among the flavan-3-ols, peak 1 ([M-H]^−^ at *m*/*z* 289; MS^2^ fragment at *m*/*z* 245) was identified as (+)-catechin by comparison of the UV spectrum and retention time with a commercial standard, and peak 4 was tentatively identified as catechin–pentoside ([M-H]^−^ at *m*/*z* 421; MS^2^ fragment at *m*/*z* 289), representing a loss of pentoside sugar unit (−132 *u*). Regarding Shen et al. [23], catechins are predominant in *Acacia catrechu*. Peak 2, the only phenolic acid found in the different extracts, was identified as sinapic acid hexoside, containing a phenolic structure attached to a hexosyl moiety, which was characterized by the pseudomolecular ion [M-H]^−^ at *m*/*z* 385 and an MS^2^ fragment at *m*/*z* 223. Peaks 3, 6, and 7 are part of a subgroup of condensed tannins composed of (+)-catechin and (-)-epicatechin units. They were tentatively identified as procyanidins and their polymer forms. Peak 3 presented pseudomolecular ions corresponding to procyanidin dimers ([M-H]^−^ at *m*/*z* 577), and peaks 6 and 7 corresponded to procyanidin trimers ([M-H]^−^ at *m*/*z* 865). 

Peak 5 ([M-H]^−^ at *m*/*z* 449) and 13 ([M-H]^−^ at *m*/*z* 595) are derivatives of eriodictyol (a flavanone), releasing a fragment at *m*/*z* 287 [eriodictyol-H]^-^, which were tentatively identified as eriodictyol-*O*-hexoside (−162 *u*, loss of a hexosyl moiety) and eriodictyol-*O*-hexosyl-deoxyhexoside (−308 *u*, loss of a hexosyl-deoxyhexoside moiety), respectively. 

In the flavonol class, peaks 8 ([M-H]^−^ at *m*/*z* 771), 9, 11 ([M-H]^−^ at *m*/*z* 625), 12 ([M-H]^−^ at *m*/*z* 479), and 18 ([M-H]^−^ at *m*/*z* 463) are derived from myricetin [24]. Their identities were assigned based on their pseudomolecular ions and MS^2^ spectra, releasing fragments corresponding to myricetin (*m*/*z* at 317) and to distinct losses of deoxyhexoside-(hexosyl-deoxyhexoside) (−146 *u* and −308 *u*), hexosyl-deoxyhexoside (−308 *u*), glucoside (−162 *u*), and rhamnoside (−146 *u*) moieties, as myricetin-*O*-deoxyhexoside-*O*-hexosyl-deoxyhexoside, myricetin-*O*-hexosyl-deoxyhexoside, myricetin-3-*O*-glucoside, and myricetin-3-*O*-rhamnoside, respectively.

Peaks 10, 14, 15, 16, 17, and 19 are derived from quercetin (max around 350–360 nm, and MS^2^ fragment at *m*/*z* 301) [25]. Peak 10 corresponds to quercetin-*O*-deoxyhexosyl-(hexosyl-deoxyhexoside) ([M-H]^−^ at *m*/*z* 755), loss of a deoxyhexosyl-(hexosyl-deoxyhexoside) (−146 *u* and −308 *u*) moiety. Peak 14 was tentatively identified as quercetin-*O*-hexosyl-deoxyhexoside ([M-H]^−^ at *m*/*z* 609). Peak 15 and 16 were positively identified by comparison of the UV spectrum and retention time with a commercial standard as quercetin-3-*O*-rutinoside ([M-H]^−^ at *m*/*z* 609) and quercetin-3-*O*-glucoside ([M-H]^−^ at *m*/*z* 463), respectively. Peak 17 ([M-H]^−^ at *m*/*z* 579), with a loss of −146 *u* and −132 *u*, was identified as quercetin-deoxyhexosyl-pentoside. Peak 19 quercetin di-deoxyhexoside ([M-H]^−^ at *m*/*z* 593) releasing an MS^2^ fragment at *m*/*z* 301 ([M-H-146-146]^−^, quercetin loss of a two deoxyhexosyl moieties.

Lastly, two flavones were tentatively identified, both derived from luteolin (max around 334 nm, and MS^2^ fragment at *m*/*z* 285). Peak 20 ([M-H]^−^ at *m*/*z* 593) and peak 21 ([M-H]^−^ at *m*/*z* 447) were identified as luteolin-7-*O*-rutinoside and luteolin-7-*O*-glucoside, respectively [26]. 

Like cistus, acacia is also rich in phenolic compounds that have already been widely explored, with 6 of the 21 tentatively identified compounds being derived from quercetin, a flavonoid with a wide range of pharmacologically beneficial activities (e.g., antioxidant, antimicrobial, anti-inflammatory, antiviral, anticancer). It has been shown to be a potential therapeutic agent for various inflammatory conditions that inhibits the proliferation of cancer cells and plays an important role in reducing blood pressure and cholesterol levels [27]. Of the 21 compounds, 5 were tentatively identified as derived from myricetin, with its consumption associated with health benefits (e.g., antioxidant and anti-neuroinflammation) and, therefore, was widely explored for the treatment of neurodegenerative diseases [28]. 

In this study, lemon verbena, while exhibiting the lowest phenolic compound diversity among the three natural matrices analyzed, still unveiled a unique and valuable array of phytochemicals. A total of nine compounds were tentatively identified, encompassing eight phenylpropanoid glycosides and one flavonol *O*-methylated [29]. The class of phenylpropanoids is characterized by their phenolic structure, which is derived from the phenylpropane pathway. Peak 1 ([M-H]^−^ at *m*/*z* 637) was tentatively identified as plantainoside C [30]. It exhibits a pseudomolecular ion [M-H]^−^ at *m*/*z* 637 and displays specific fragmentation patterns in its mass spectrometry, including characteristic ions in MS^2^ at *m*/*z* 351, 285, 193, and 175 [31]. Peaks 2, 3, 7, and 8 are tentatively identified as various verbascoside derivatives. Being tentatively identified according to a previously reported elution order by Leyva-Jiménez et al. [32]. Peak 2 and 3 ([M-H]^−^ at *m*/*z* 639) were identified as β-hydroxy-verbascoside and β-hydroxy-isoverbascoside, respectively. Peaks 7 and 8 were confidently determined as verbascoside and isoverbascoside, respectively, featuring a pseudomolecular ion [M-H]^−^ at *m*/*z* 623. Martynoside, with a pseudomolecular ion [M-H]^−^ at *m*/*z* 651, was tentatively identified as peak 4, adding to the phenylpropanoid diversity. Peak 5 was identified as forsythiaside, and peak 11 as eukovoside, with respective pseudomolecular ions of [M-H]^−^ at *m*/*z* 623 and 637. Regarding the flavonol, with a pseudomolecular ion [M-H]^−^ at *m*/*z* 491, peak 6 was identified as isorhamnetin-3-*O*-glucuronide. 

Leyva-Jiménez et al. studied the optimization of phenylpropanoids and flavonoids from lemon verbena leaves and detected a total of 38 compounds. This quantitative difference in the variety of compounds present in the extracts is probably due to the fact that the authors used a different extraction technique (supercritical fluid system) that promoted the extraction of lipophilic phenolic compounds primarily [29]. Phenolic compounds are important natural compounds with relevant biological activities. Different lemon verbena extracts have been subject to bioactivity studies, which have demonstrated their antibacterial, antioxidant, and anti-inflationary activities, among others [11]. In this work, four of the nine compounds were tentatively identified as derivatives of verbascoside, a phenylethanoid glycoside, with various pharmacological properties, such as antioxidant, antibacterial, anti-inflammatory, antineoplastic, anti-androgenic, wound-healing, and neuroprotective properties. The combination of some of its pharmacological effects is promising for the development of pharmacological treatments for acne vulgaris [33].

### 2.2. RSM Conditions and Optimized Responses

Mathematical tools such as RSM have helped reduce the number of experimental runs in order to optimize processes. Using these tools, the operational costs are lower, and the efficiency of resources is higher [13]. 

In this work, through RSM tools, six extracts rich in phenolic compounds were obtained, which can be used for future applications as pharmaceutical and nutraceutical compounds. Using the Box–Behnken design, the optimization procedure included 29 individual extractions, with four independent variables and obtained as response 1 (R_1_) the maximization of the extraction of the most abundant phenolic compound and as response 2 (R_2_) the maximization of the extraction yield. The upper and lower limits of each factor were decided after careful consideration of previously published results in the literature [2,34,35,36,37,38]. 

Table 4 shows the responses obtained for each sample prior to the optimization procedures. Results are expressed in mg of major phenolic compound per g of extract (mg/g dw) for R_1_ and in g of extraction yield per g (g/g dw) for R_2_. In cistus, the most common polyphenol group in both extraction systems was the group of hydrosoluble tannins, such as punicalagin. For this optimization, the total concentration of these compounds, peaks 1, 2, 3, 6, 7, 13, 14, and 15 were taken into consideration. In acacia, the condensed tannins, such as procyanidin dimers and trimers (peaks 3, 6, and 7) for the HAE methodology were the response factors; thus, for UAE, the different myricetin glycosides (peaks 8, 9, 11, 12, and 18) were the main compounds applied to the optimization of this extraction system. Finally, for lemon verbena, martynoside (peak 4) was the most abundant compound in HAE and used for this response, whereas for UAE, forsythiaside (peak 5) was the most abundant compound and, therefore, applied as the response factor for this extraction methodology. 

The coded equation for each sample and each optimization is shown in Table 5, in which the contribution of each factor can be understood. For the RSM, all analyses were performed at a confidence interval of α = 0.05, excluding non-significant values. Regarding each coded equation, *X*_1_, *X*_2_, *X*_3_, and *X*_4_ represent each factor individually, as well as their contribution, meaning that the higher the value next to each of those factors, the more influence that factor has on the outcome. X_x_X_y_ represents the interactions between two factors, while X_x_^2^ represents the squared value of each factor. 

Concerning each coded equation, very consistent results were found for each extraction method. Of all the factors in both extraction types, the percentage of ethanol was the most important factor, followed by time and temperature. Power, in watts (W), was the only most important factor for the two extractions, while the solid-to-liquid ratio was the least important in all extraction types. Interestingly, in all plant species, and considering both extraction types, for R_1_ (major phenolics), the factor with the highest influence shows a considerable difference from the others, while for R_2_ (extraction yield), the two most important factors are quite close in terms of influence on the outcome. These results show that for obtaining individual bioactive components, one factor stands out, namely, ethanol, while for the extraction yield, a combination of two factors (ethanol and temperature) promotes the highest yield. In terms of consistencies over the different plant species, extraction temperature is the most influential in all plants using heat-assisted extraction to obtain the highest extraction yield. Considering HAE and R, for both cistus and verbena, extraction time was the most important factor, while for acacia, it was ethanol percentage. For UAE, considering R_1_ once again, ultrasonic power was the most important for verbena and acacia, while extraction time was the most influential for cistus. Finally, for UAE, ethanol was the most important factor for the extraction yield. Overall, more consistency was found in terms of the factors across extraction types, while for each plant, the most important factor varied.

The variation in the most important factor justifies the need for optimization studies, as different techniques may have higher yields at lower costs in terms of time and reagents. 

The results of the optimal extraction conditions (predicted condition at which the maximum amount of the major polyphenols or extraction yield is found), as well as the optimal individual and global (maximization of the extraction yield and major phenolic compound) responses, are shown in Table 6, following a discussion considering each species. Overall, considering extraction yield, the plant with the most quantity was lemon verbena, with 0.39 mg/g obtained through HAE. The optimal point was set at 31% ethanol, 120 min of extraction time, and 80 °C and 40 g/L, while the lowest extraction yield was recorded for acacia obtained through UAE at an optimal point of 70% ethanol, 20 min, and 483 W and 10 g/L. These findings show that in terms of compounds, lemon verbena has a higher quantity when compared to cistus and acacia, the latter being the plant with the least amount. Considering the major phenolic, the plant that showed the highest amount was cistus, extracted through UAE, predicting a maximum of 284 mg/g of total punicalagins, with an optimal point at 3% ethanol, 22 min of extraction time, 181 W, and 35 g/L. Inversely, and at the maximum extraction point, the plant with the lowest amount of total forsythiaside (major phenolic compound) was verbena extracted through UAE, with an optimal point of 94% ethanol, 25 min of extraction time, 399 W, and 29 g/L. Although the major phenolic compound was compared, it should be pointed out that these values also depend on the real amount of the major compound in each plant. Derringer’s desirability function is also present in Table 6; this function allows for the identification of the experimental conditions that show the optimal levels for all evaluated variables. Thus, if the major compound is relatively high, there should not be a major change in the optimal points between the major compound and the desirability and/or a reduction in the predicted amount of the major compound. Considering the three plants, the lowest difference between desirability and the two responses was for cistus extracted with HAE, where the difference between total punicalagins was only 4 mg/g, with no changes in extraction yield. Inversely, the highest difference was also recorded for Cistus but extracted through UAE, where the difference between the optimal value and the desirability of total punicalagins was almost 40 mg/g. 

#### 2.2.1. Optimization of the Extraction Parameters of Cistus

For R_1_ (TPu) and R_2_ (DR) of the cistus extracted through HAE, after analysis of the 29 experimental runs, quadratic functions were defined for each, with a significant model and a non-significant lack of fit, with an adjusted R^2^ of 0.8267 (R_1_) and 0.9479 (R_2_), and coded Equations (1) and (2), respectively (Table 5). Thus, the optimum values of the independent variables that maximized the number of extractable punicalagins were set at 29% ethanol, 178 min, 30 °C, and an S/L ratio of 11 g/L, which are previewed to render 266 mg/g. To maximize the extraction yield, the variables were set at 48% ethanol, 121 min, 80 °C, and 36 g/L for the S/L ratio, resulting in 0.35 g/g. Using the desirability function considering both responses, the optimal conditions to maximize the extraction of punicalagins and extraction yield simultaneously were determined, and the following values were obtained: ethanol (%, *v*/*v*) = 34; extraction time (min) = 120; temperature (°C) = 80; S/L ratio = 10, allowing for the extraction of 262 mg/g punicalagins and 0.35 g/g extraction yield.

The final step in terms of optimization is to understand how each factor varies over the defined interval. Thus, in Table 7, the different 3D plots for HAE of cistus are shown. The 3D plots show how two factors vary at a time, with the other two fixed at the optimal point. In general, R_1_ shows that the maximization of punicalagin extraction is not significantly influenced by temperature and S/L ratio, but it is favored by the low ethanol contents (<40%) and lightly favored by an extraction time of more than 110 min. In the extraction of R_2_, the independent variable time has a low influence, while the S/L ratio has no relevant influence on its maximization, which occurs at high temperatures (>50 °C) and at ethanol percentages between 20 and 80%. This is quite evident due to the lack of a surface area, for instance, in the temperature vs. ratio chart of R_2_, where, rather than a curve, a plane is formed. The charts for the desirability function demonstrate the appropriate conditions for the joint optimization of the two responses under study, as well as individually; it was found that the S/L ratio has a low influence on the joint maximization of both responses and time and temperature prove to have little influence; however, higher temperatures promote the extraction of punicalagins and extraction yield at the same time, with the percentage of ethanol (5% to 60%) being the determining variable for optimizing the extraction of both factors. 

In the UAE analysis of the 29 runs, quadratic functions were also sought, although four runs were eliminated as outliers (in R_1_ and R_2_), resulting in a non-significant lack of fit and an adequate fit of the model, with an adjusted R^2^ of 0.7062 (R_1_) and 0.9701 (R_2_), described by Equations (3) (R_1_) and 4 (R_2_), respectively. Thus, the maximization of the punicalagin concentration occurs under the conditions of 3.22% ethanol, 22 min, 171 W, and 35 g/L, giving 284 mg/g; the extraction yield is maximized under the conditions of 55% ethanol, 27 min, 421 W, and 39 g/L, resulting in 0.35 g/g; the joint maximization of the two responses occurs when the conditions of 37% ethanol, 30 min, 406 W, and 25 g/L are applied, giving 246 mg/g (R_1_) and 0.33 g/g (R_2_). The optimal values are presented in Table 8, where the first column refers to R_1_, the second column to *R*_2_, and the third column to the desirability function. In the 3D charts, the percentage of ethanol (0–50%) and the S/L ratio (>30 g/L) are the independent variables that have the greatest influence on the optimization of punicalagin extraction. It was also found that lower ultrasonic powers favor the extraction of these compounds. With respect to R_2_, both the percentage of ethanol and the ultrasonic power applied have a weak effect on the variation in the extraction yield obtained, in contrast to the S/L ratio (>40 g/L) and the extraction time (>25 min), which proved to be decisive for the optimization of this factor. When analyzing the charts obtained for the desirability function, none of the independent variables had a higher weight in the joint optimization of both responses; however, it is noted that higher values of ultrasound power favor the extraction of both responses. 

#### 2.2.2. Optimization of the Extraction of Acacia

For R_1_ of the acacia sample obtained by HAE extraction, analysis of the 29 runs yielded the quadratic Equation (5); after eliminating five runs that were considered outliers, a significant model with a non-significant lack of fit was obtained, with an adjusted R^2^ of 0.6118. The function of maximizing the extraction of total procyanidin led to an optimum point of 74% ethanol, 86 min, 24 °C, and 50 g/L, obtaining 233 mg/g. For R_2_, the quadratic Equation (6) was determined, eliminating three outlier runs, resulting in a significant model and a non-significant lack of fit, with an adjusted R^2^ of 0.9651. Therefore, an optimum point (0.37 g/g) is predicted when extractions with 28% ethanol, 120 min, and 80 °C and 12 g/L are applied. Using the desirability function considering the two responses, 69% ethanol, 173 min, 73 °C, and 11 g/L, the values of the independent variables were set to maximize the extraction of procyanidins and extraction yield, resulting in 287 mg/g procyanidins and 0.34 g/g extraction yield. Table 9 shows the 3D response surface charts for R_1_ (first column), R_2_ (second column), and the desirability function (third column) resulting from the HAE. Briefly, the most important factors for maximizing the extraction of procyanidins (R_1_) were temperature (<40 °C) and S/L ratio; however, the interaction between extraction time and solvent showed a wider range of values, maximizing the extraction of procyanidins. In terms of R_2_*,* the extraction time (>110 min) is the variable with the greatest influence on the optimization of the extraction yield, in contrast to what happens with the percentage of ethanol. The analysis of the charts resulting from the desirability function shows that time and temperature are the variables with the greatest influence on the equilibrium of each response, favoring the maximization of the two extraction responses with long times and high temperatures. 

For the UAE technique, analysis of the 29 runs yielded in the quadratic Equation (7) for R_1_ after eliminating one outlier and the quadratic Equation (8) for R_2_ after eliminating four outlier runs, which yielded a non-significant lack of fit and an adequate fit of the model for both responses. This resulted in an adjusted R^2^ of 0.8024 for R_1_ and 0.9623 for R_2_. Thus, to maximize myricetin extraction (111 mg/g), the optimal values were set at 65% ethanol, 8 min, 50 W, and 50 g/L; to optimize the extraction yield (0.24 g/g), the variable values were set at 70% ethanol, 20 min, 483 W, and 10 g/L; and finally, for the desirability function, the values were set at 65% ethanol, 17 min, 500 W, and 10 g/L, resulting in 105 mg/g myricetin and 0.23 g/g residue. The 3D response surface charts presented in Table 10 are the result of the interaction between the variables used in the UAE method. Briefly, for R_1_, ethanol percentage (40–90%) proved to be the variable with the greatest weight in maximizing myricetin extraction, with time being the variable with the least influence; for R_2_, the power (>350 W) is the variable with the most influence on obtaining a greater dry weight yield, as opposed to the S/L ratio; as for the simultaneous maximization of the two responses, both the high ethanol percentage (50–100%) and the high powers (>410 W) favor the optimization of the responses.

#### 2.2.3. Optimization of the Extraction of Lemon verbena

For R_1_, from the extractions of lemon verbena obtained by HAE, the examination of the 29 runs, after removing two outlier runs, yielded a quadratic function with a significant model, with a non-significant lack of fit and with an adjusted R^2^ of 0.9342, as confirmed in the coded Equation (9). For R_2_, the values also allowed for obtaining a significant model, a non-significant lack of fit, and an adjusted R^2^ of 0.9632, with the coded Equation (10). Thus, the optimal values for maximizing the amount of martynoside were set at 13% ethanol, 96 min, 49 °C, and 17 g/L, which predicted an extraction of 114 mg/g; the optimal values for maximizing the amount of extraction yield (0.39 g/g) were 31% ethanol, 120 min, 80 °C, and 40 g/L; after applying the desirability function, the values of the independent variables were fixed, to maximize martynoside (109 mg/g) and extraction yield (0.34 g/g) in 30% ethanol, 120 min, 64 °C, and 10 g/L. Table 11 shows the 3D surface charts resulting from the different interactions between the variables (from two to two) in the application of HAE. Column one represents the charts of *R*_1_; column two represents those of *R*_2_, and column three represents the charts resulting from the application of the desirability function. For the optimization of *R*_1_, the percentage of ethanol is the most important variable, suggesting that values between 20 and 50% ethanol and lower temperatures favor the extraction of martynoside, in contrast to what was observed for extraction time and S/L ratio. For R_2_, as well as for R_1_, the solvent (0–60%) is the variable with the greatest weight in the optimization of this factor, and temperature also demonstrates an influence on the optimization of R_2_, but in contrast to what happens for R_1_, in this case, when higher temperatures favor residue optimization. As for R_1_ and R_2_, for the desirability function, solvent (0–60% ethanol) is, in this case, the crucial variable for maximizing the extraction of martynoside and extraction yield. 

Regarding the optimization of extraction using the UAE technique, the execution of the 29 runs resulted in the coded Equation (11) for *R*_1_ and (12) R_2_. Significant models with a non-significant lack of fit were obtained for both responses after removing one outlier run for R_1_ and three for R_2_, resulting in an R^2^ of 0.9693 (R_1_) and 0.9405 (R_2_). To optimize extraction, the variable values were set to 94% ethanol, 25 min, 399 W, and 29 g/L to maximize extraction of forsythiaside (101 mg/g); 7% ethanol, 12 min, 387 W, and 38 g/L were set to maximize the residue (0.31 g/g); and finally, 66% ethanol, 30 min, 400 W, and 10 g/L were set for simultaneous maximization of forsythiaside (96 mg/g) and extraction yield (0.26 g/g). Although the R_1_ optimization of lemon verbena extraction varies according to the extraction method used, it is clear from the analysis of the 3D surface plots that even in R_1_ extraction optimization with the application of UAE (Table 12), the variable with the greatest influence on maximization is the percentage of ethanol (>70%) and although the application of high potentials seems to favor the extraction of forsythiaside, the extraction time and the S/L ratio are not considered important factors for the desired effect. For R_2_, gaining the ethanol percentage is crucial for maximizing the extraction yield, but in contrast to R_1_, lower ethanol percentages (<50%) favor the extraction of high residue amounts in this case. This response is also influenced by power (>300 W), and again, both the variation in the extraction time and the S/L ratio are not relevant for optimizing the reaction. When using the desirability function, the simultaneous maximization of R_1_ and R_2_ is mainly influenced by the variable ethanol content (40–90%) and the variable power (>400 W).

#### 2.2.4. General Considerations

In cistus, both conventional extraction (maceration) and green extraction (assisted by ultrasound) extracted the punicalagins in larger quantities. These compounds belong to the class of plant ellagitannins, which are frequently found in pomegranates. The punicalagins are scientifically associated with various biological effects, particularly antibacterial, antioxidant, hepatoprotective, antidiabetic, and anti-obesity activities, highlighting the potential of these compounds for use in the pharmaceutical and nutraceutical industries [39]. Xu et al. investigated the potential of punicalagins as drugs for the treatment of Alzheimer’s disease by combining simulations of molecular dynamics with animal experiments. The authors demonstrated that punicalagins have a high pharmacological potential for the prevention of Alzheimer’s disease, as their mechanisms of action counteract the deposition of neurotoxic proteins and improve cognitive damage [40]. In general, for both extraction techniques studied, the percentage of ethanol is the variable that mostly affects the extraction of punicalagins, although maximization occurs when solvents with low ethanol percentages are used. According to the obtained coded equations, the ultrasound-assisted extraction method proves to be the most sustainable process for obtaining extracts rich in punicalagins since it involves the use of a lower amount of ethanol, low ultrasound powers, and short extraction times; although it implies the use of a greater amount of material, a greater quantity of punicalagins is also obtained. The high value of the extraction yield does not mean that a large number of punicalagins were extracted, but the extraction yield may indicate a tendency toward a higher content of bioactive compounds. In this study, we found that the amount of extraction yield obtained is influenced by different factors depending on the extraction method. For HAE, the amount of ethanol contained in the solvent is the most important variable, while for UAE, although high ultrasonic powers favor the amount of residue obtained, the extraction time and S/L ratio have a greater influence on the amount of residue obtained. The results obtained suggest that the UAE method will be of greater interest on an industrial scale. Although the extraction yield does not differ between the two methodologies applied, the UAE method uses much shorter extraction times, with a slightly different S/L ratio and ethanol percentage, compared to the HAE method, indicating lower energy costs for the industry. The optimization of the extraction of punicalagins and extraction yield is simultaneously influenced by the four variables, with the percentage of solvent being the key variable when HAE is applied and the power variable having the greatest weight when UAE is applied. Although HAE predicts a higher yield of punicalagins and extraction yield, UAE is more sustainable and practical for the industry as shorter extraction times are applied. Živković et al. performed an optimization study on the extraction of compounds from pomegranate peels, in which they maximized the extraction of punicalagins; the authors set the optimal conditions at 35 min, 50% ethanol, an S/L ratio of 1:30, and a temperature of 20 °C, at which they recovered 35.05 mg/g dw, a value well below the one predicted in this work for both heat-assisted extraction and ultrasound-assisted extraction [41]. To the author’s knowledge, this study is a pioneer in optimizing the extraction of punicalagins from cistus. With regard to acacia, the application of HAE favors the extraction of procyanidin, a polyphenol that is most abundant in plants and has been extensively studied. Scientifically associated with pharmacological activities such as antioxidant, antibacterial, anticancer, and anti-inflammatory effects, it also has an important role in protection against chronic diseases, and its beneficial effect in the treatment of Alzheimer’s disease has also been reported [42]. The use of UAE favors the extraction of different myricetin glycosides, a flavonoid produced in the growth process of the plant that has a high pharmacological potential. Several preclinical pharmacological activities have been attributed to this compound, which has been demonstrated in Alzheimer’s, Parkinson’s, and Huntington’s diseases. It also plays an important pharmacological role in antidiabetic, analgesic, antihypertensive, and immunomodulatory effects [43]. The difference in most extracted compounds is due to the extraction conditions applied since both temperature and ultrasonic power are factors that can cause the degradation of compounds, and the percentage of ethanol can favor the extraction of a certain group of compounds. In this case, temperature affects the extraction of procyanidins, and a high percentage of ethanol favors the extraction of myricetin. When the feasibility of the two extractions is reviewed, it can be concluded that both are feasible at an industrial level, depending on the final objective, but HAE promotes a higher extraction yield compared to UAE. As for the extraction yield, temperatures (HAE) and high ultrasonic powers (UAE) increase the yield; however, although extraction by maceration is expected to have a higher yield, this method requires the application of high times and temperatures, which may reduce its industrial interest. Consequently, when observing the results of optimizing the two responses at the same time, HAE also achieves better yields but requires high times and temperatures, which can become an unsustainable process. To the authors’ knowledge, no work has yet been published on optimizing the extraction of procyanidins and different myricetin glycosides from acacia leaves. These two compounds, naturally occurring in several plants, have already undergone extraction optimization; for example, Tomaz et al. optimized their extraction from grape skins and predicted responses (procyanidins—0.12 mg/g dw; myricetin—0.62 mg/g dw) much lower than those obtained in this work [44], thus demonstrating the potential of acacia leaves for the development of ingredients rich in procyanidins and myricetin that could have applications in various areas, particularly in pharmaceutical area. As with acacia, lemon verbena has an extraction methodology that favors the extraction of different groups of compounds. In HAE, an extract rich in martynoside was obtained, a phenylpropanoid glycoside that is highly studied in Chinese medicine for its pharmacological properties (e.g., antagonizing sports anemia, delayed skeletal muscle fatigue, functioning as a new natural selective modulator of the estrogen receptor) [45]. In the UAE, the extracts containing high amounts of forsythiaside, a phenylethanoid glycoside that, like martynoside, has been widely explored in Chinese medicine and is associated with various biological activities, such as antibacterial, antiviral, antioxidant, vasodilatory, and neuroprotective effects [46]. In general, the percentage of ethanol in the solvent affects the extraction yield of the two responses for the two extraction methods used; in short, for HAE, an ethanol percentage of less than 60% increases the extractability of the compounds; for UAE, an ethanol percentage greater than 70% increases the extractability of forsythiaside but negatively influences the amount of residue obtained. Choosing one extraction methodology over the other is only possible if both optimize the same group of compounds, which is not the case in this study. However, after considering the coded equations, it is concluded that the UAE technique can be more sustainable for all the responses studied since it applies high powers, but the extraction times are shorter, and it does not use temperature. More sustainable technologies are trending and are at the top of the list of global demands, which is why it is so important to offer the industry attractive and economically viable alternatives. Once again, as far as it is known, there are no studies that optimize the extraction of martynoside and forsythiaside from lemon verbena; however, Leyva-Jiménez et al. have identified and quantified phenolic compounds from the leaves of lemon verbena, and although units are not the same, after conversion (martynoside—0.807 mg/g; forsythiaside—0.466 mg/g), the advantages of optimizing extraction processes become clear [47].

## 3. Materials and Methods

### 3.1. Samples

Of the three wild plants, fresh leaves of flowering *Cistus ladanifer* L. (cistus) and *Acacia dealbata* L. (acacia) were collected at random from wild plants in Bragança (decimal degrees, 41.789653, −6.757697; 41.802676, −6.756686), Portugal, in February 2021. The identification of the plants was based on the herbarium Escola Superior de Bragança—Instituto Politécnico de Bragança, which uses morphological characteristics from the Iberian Flora to correctly characterize the species. After harvesting, the plant material was dried at room temperature and protected from light. The dried *Aloysia citrodora* Paláu (Lemon verbena) was provided by an organic farmer (Cantinho das Aromáticas, Vila Nova de Gaia, Portugal). All matrices have been reduced to a fine-dried powder (∼20 mesh) and stored protected from light and heat until further analysis. 

#### 3.1.1. Standards and Reagents

The ethanol for extraction was acquired from a scientific retailer and was of P.A. purity. The solvents for Ultra High-Performance Liquid Chromatography (UPLC), were of HPLC grade. The ethanol (HPLC) and acetonitrile (HPLC) were acquired from Fisher Scientific (Lisbon, Portugal). Formic acid was acquired from Sigma-Aldrich (St. Louis, MO, USA). Water was treated in a Milli-Q water purification system (TGI Pure Water Systems, Greenville, SC, USA). The phenolic compounds standards were from Extrasynthese (Genay, France). 

#### 3.1.2. Heat-Assisted Extraction (Dynamic Maceration)

The heat-assisted extraction (dynamic maceration) was carried out according to the modified methodology described by Leichtweis et al. [13]. For this purpose, a thermostatic bath was used with stirring performed with submersible magnetic stirrers (Micro Stirrers, Thermo Scientific Cimarec, Thermo Fisher Scientific, Waltham, MA, USA). An amount of 20 mL of solvent was added to a dried sample, in which the percentage of solvent (ethanol) and masses of the sample were added to different extraction cups Table 13, Table 14 and Table 15. Temperature, extraction time, and S/L ratio were also varied for each extraction. After the extraction process, the ethanolic phase of the mixtures was evaporated at 40 °C using a rotary evaporator (Büchi R-210, Flawil, Switzerland), and the aqueous phase was freeze-dried (FreeZone 4.5, Labconco, Kansas City, MO, USA) and stored at room temperature protected from light until further analysis. The dry weight of each extraction was calculated by drying 5 mL of each sample at 100 °C for 3 days after the extraction process. The results throughout this manuscript are expressed in grams of residue per gram of dry weight (g/g dw). 

#### 3.1.3. Ultrasound-Assisted Extraction (UAE)

Ultrasound-assisted extractions were performed according to the modified procedure described by Leichtweis et al. [13]. The known amount of dry sample powder was mixed with 50 mL of solvent and processed according to the experimental design matrix (Table 13, Table 14 and Table 15) in an ultrasonic device (QSonica Sonicators, model CL-334, Newtown, CT, USA) containing a fixed water reactor at a frequency of 40 kHz. Extractions were performed at room temperature and in an ice bath to maintain the temperature constant. The varying conditions were the ultrasonic power, extraction time, solvent (ethanol) percentage, and solid-to-liquid ratio. The dry weight of each extraction was calculated by drying 5 mL of each sample at 100 °C for 3 days after the extraction process. The results throughout this manuscript are expressed in grams of residue per gram of dry weight (g/g dw).

#### 3.1.4. Analysis of Phenolic Compounds

The analysis of the phenolic composition of the different extracts was carried out according to the modified procedure described by Mandim et al. [48]. An amount of 10 mg of the different extracts was redissolved in 1 mL of ethanol:water (20:80, *v*/*v*) and filtered through a syringe filter (0.22 µm). Chromatographic analysis was performed by Ultra High-Performance Liquid Chromatography (UPLC Dionex Ultimate 3000 Thermo Scientific, San Jose, CA, USA) coupled to a diode array detector (280, 330 and 370 nm) and ionization mass spectrometry by electrospray (Linear Ion Trap LTQ XL, Thermo Finnigan, San Jose, CA, USA) (UPLC-DAD-ESI/MS). Chromatographic separation of compounds was performed at 35 °C on a Waters Spherisorb S3 ODS-2 C18 column (3 µm, 4.6 mm × 150 mm, Waters, Milford, MA, USA). An attempt to identify compounds was made by comparing their retention times (UV-Vis and mass spectra) with the available standards and literature information. Calibration curves of phenolic standards were carried out based on the UV-Vis signal to obtain a quantification of the phenolic compounds, when commercial standards are unavailable; the quantification was performed with the calibration curve of the most similar standard ((+)-catechin (y = 84,950x − 23,200, R^2^ = 1; LOD = 0.17 μg/mL; LOQ = 0.68 μg/mL), quercetin 3-*O*-glucoside (y = 34,843x − 160,173; R^2^ = 0.9998; LOD = 0.21 µg/mL; LOQ = 0.71 µg/mL), hydroxybenzoic acid (y = 20,800x + 41,309, R^2^ = 0.9986; LOD = 0.41 µg/mL; LOQ = 1.22 µg/mL), apigenin 6-*C*-glucoside (y = 107,025x − 61,531; R^2^ = 0.9989; LOD = 0.19 µg/mL; LOQ = 0.63 µg/mL), gallic acid (y = 131,538x + 292,163, R^2^ = 0.9969, LOD = 8.05 μg/mL and LOQ = 24.41 μg/mL), ellagic acid (y = 26,719x − 317,255, R^2^ = 0.9986; LOD = 0.41 µg/mL and LOQ = 1.22 µg/mL), caffeic acid (y = 388,345x + 406,369; R^2^ = 0.994; LOD = 0.78 μg/mL; LOQ = 1.97 μg/mL), sinapic acid (y = 197.337x + 30,036; R^2^ = 0.999; LOD = 0.17 μg/mL; LOQ = 1.22 μg/mL), naringenin (y = 18,433x + 78,903, R^2^ = 0.9998; LOD = 0.20 μg/mL; LOQ = 0.64 μg/mL), and Verbascoside (y = 124.233x − 18.873, R^2^ = 1; LOD = 0.70 µg/mL and LOQ = 2.13 µg/mL)). The results were expressed in milligrams of major phenolic compounds per g of extract of dry weight (mg/g dw). 

#### 3.1.5. Experimental Design and Statistical Analysis 

The experimental project was designed following the modified methodology described by Leichtweis et al. [13]. Design Expert 8.0.6 software (State-Ease Inc., Minneapolis, MN, USA) was used to design and interpret the results. The Box–Behnken design (BBD) was used, with three levels of each of the four factors trialed to obtain the optimal response. In the case of maceration extraction (HAE), the varying factors were solvent (ethanol) percentage, which varied between 0 and 100% (*X*_1_), extraction time (5 to 180 min (*X*_2_)), extraction temperature (20 to 80 °C (*X*_3_)), and solid to liquid ratio (10 to 50 g/L (*X*_4_)). The ultrasound-assisted extraction also had four factors varying, namely, the ultrasonic power (50 to 500 W (*X*_1_)), extraction time (5 to 30 min (*X*_2_)), solvent (ethanol) percentage (0 to 100% (*X*_3_)), and solid to liquid ration (10 to 30 g/L (*X*_4_)). In both extraction types, the optimization was performed for two responses, namely, R_1_—major phenolic compound (in which the most abundant phenolic compound was different from sample to sample) and R_2_—extraction yield. It was also determined that desirability represents the optimal point to maximize both responses (extraction yield and major phenolic compound). Table 13, Table 14 and Table 15 show the various experimental designs, including the four factors and the values for each of the 29 different extractions performed. Using the base quadratic Equation (13), the mathematical prediction models were built, where Rn are the responses, and bn are the intercept, linear, quadratic, and interaction terms of each of the four factors.
R_n_ = b_0_ + b_1_*X*_1_ + b_2_*X*_2 +_ b_3_*X*_3_ + b_4_*X*_4_ + b_12_*X*_1_*X*_2_ + b_13_*X*_1_*X*_3_ + b_14_*X*_1_*X*_4_ + b_23_*X*_2_*X*_3_ + b_24_*X*_2_*X*_4_ + b_34_*X*_3_*X*_4_ + b_11_*X*_1_^2^ + b_22_*X*_2_^2^ + b_33_*X*_3_^2^+ b_44_*X*_4_^2^(13)

An analysis of variance (ANOVA) was used to determine the coefficient of determination (R^2^) of the model equations, and the F test was used to test the significance value (α = 0.05).

## 4. Conclusions

The pharmacological potential of phenolic compounds has been explored by various authors who demonstrated that these compounds were a potential substitute for bioactive agents in pharmacy to promote human health and to prevent and cure various diseases. 

In this work, three wild plants were investigated as sources of compounds with proven bioactivity, including two widespread invasive plants (cistus and acacia) and an aromatic plant (lemon verbena). For rockrose, 15 phenolic compounds were tentatively identified; both extractions used promoted the extractability of the punicalagins; UAE proved to be the most efficient extraction method. For acacia, 21 phenolic compounds were tentatively identified; HAE promotes the extraction of procyanidin dimers and trimers, and UAE favors the extraction of various myricetin glucosides; HAE favors a higher extraction yield of phenolic compounds. For lemon verbena, nine phenolic compounds were tentatively identified; HAE favors the extraction of martynoside, and UAE favors the extraction of forsythiaside; with the application of HAE, a higher extraction yield of phenolic compounds is obtained. In general, it has been found that the percentage of ethanol was the variable that most influenced the extraction of phenolic compounds. UAE proves to be the most economical and sustainable method.

This work provides the scientific community and industry with optimized protocols for the extraction of different phenolic compounds from plants that are under-researched for their phenolic content and allows for an improvement in extraction efficiency to contribute to the sustainability of natural ingredient development processes. Further studies are needed to demonstrate the bioactive effects and the potential of the optimal extracts obtained in this work for industrial, especially pharmaceutical, use in order to provide the industry with compounds with the pharmacological potential identified and also to contribute to the valorization of wild species.

## Figures and Tables

**Table 1 pharmaceuticals-17-00593-t001:** Retention time (Rt), wavelengths of maximum absorption in the UV-Vis region (λmax), fragmentation profiles, and tentative identification of phenolic compounds in cistus.

Peak	Rt (min)	λmax (nm)	[M-H]^−^ (*m*/*z*)	MS^2^ (*m*/*z*)	Tentative Identification
1	5.05	258, 380	1083	781 (17), 601 (16), 301 (100)	Punicalagin isomer I
2	5.92	259, 378	1251	1083 (4), 781 (13), 601 (4), 301 (13)	Punicalagin gallate isomer I
3	7.38	258, 378	1251	1083 (4), 781 (13), 601 (4), 301 (13)	Punicalagin gallate isomer II
4	10.48	338	757	595 (100), 463 (8), 301 (9)	Quercetin-*O*-di-hexose-*O*-pentose
5	13.11	342	625	301 (100)	Quercetin-dihexoside
6	14.61	259, 378	1251	1083 (4), 781 (13), 601 (4), 301 (13)	Punicalagin gallate isomer III
7	15.72	258, 380	1083	781 (17), 601 (16), 301 (100)	Punicalagin isomer II
8	17.61	341	433	301 (100)	Quercetin-*O*-pentoside
9	21.02	342	447	285 (100)	Kaempferol-3-*O*-glucoside
10	21.96	345	447	301 (100)	Quercetin-*O*-deoxyhexoside
11	23.92	334	593	285 (100)	Kaempferol-3-*O*-rutinoside
12	24.64	332	667	505 (65), 463 (41), 301 (100)	Quercetin-*O*-acetyl-di-hexoside
13	33.55	380	1083	781 (17), 601 (16), 301 (100)	Punicalagin isomer III
14	36.03	384	1251	1083 (4), 781 (13), 601 (4), 301 (13)	Punicalagin gallate isomer IV
15	36.1	384	1251	1083 (4), 781 (13), 601 (4), 301 (13)	Punicalagin gallate isomer V

**Table 2 pharmaceuticals-17-00593-t002:** Tentative identification, retention time (Rt), wavelengths of maximum absorption in the visible region (λmax), and mass spectral of phenolic compounds of the acacia.

Peak	Rt (min)	λmax (nm)	[M-H]^−^ (*m*/*z*)	MS^2^ (*m*/*z*)	Tentative Identification
1	5.92	273	289	245 (100)	(+)-Catechin
2	6.73	278	385	223 (100)	Sinapic acid hexoside
3	7.38	280	865	289 (100)	Procyanidin trimer
4	7.43	286	421	289 (100)	Catechin-pentoside
5	7.89	281	449	287 (100)	Eriodictyol-*O*-hexoside
6	8.31	280	577	289 (100)	Procyanidin dimer
7	9.79	280	577	289 (100)	Procyanidin dimer
8	10.84	256	771	625 (15), 463 (45), 317 (100)	Myricetin-*O*-deoxyhexoside-*O*-hexosyl-deoxyhexoside
9	12.08	356	625	317 (100)	Myricetin-*O*-hexosyl-deoxyhexoside
10	13.56	355	755	301 (100)	Quercetin-*O*-deoxyhexosyl- (hexosyl-deoxyhexoside)
11	14.17	356	625	317 (100)	Myricetin-*O*-hexosyl-deoxyhexoside
12	14.49	355	479	317 (100)	Myricetin-3-*O*-glucoside
13	15.12	355	595	287 (100)	Eriodictyol-*O*-hexosyl-deoxyhexoside
14	16.64	355	609	301 (100)	Quercetin-*O*-hexosyl-deoxyhexoside
15	17.34	355	609	301 (100)	Quercetin-3-*O*-rutinoside
16	17.88	350	463	301 (100)	Quercetin-3-*O*-glucoside
17	18.42	352	579	301 (100)	Quercetin-*O*-deoxyhexosyl-pentoside
18	19.96	353	463	317 (100)	Myricetin-3-*O*-rhamnoside
19	22.71	356	593	301 (100)	Quercetin-*O*-di-deoxyhexoside
20	23.45	341	593	285 (100)	Luteolin-7-*O*-rutinoside
21	24.23	341	447	285 (100)	Luteolin-7-*O*-glucoside

**Table 3 pharmaceuticals-17-00593-t003:** Tentative identification of phenolic compounds of the lemon verbena. The retention time (Rt), wavelengths of maximum absorption in the visible region (λmax), and mass spectral data are presented.

Peak	Rt (min)	λmax (nm)	[M-H]^−^ (*m*/*z*)	MS^2^ (*m*/*z*)	Tentative Identification
1	7.32	347	637	351 (43), 285 (100), 193 (39), 175 (5)	Plantainoside C
2	7.57	331	639	161 (100), 179 (47)	β-Hydroxy-verbascoside
3	7.96	330	639	161 (100), 179 (28)	β-hydroxy-Isoverbascoside
4	13.81	343	651	351 (72), 299 (5), 193 (100)	Martynoside
5	14.54	330	623	461 (11), 315 (15), 179 (5), 161 (100)	Forsythiaside
6	15.51	337	491	315 (100), 300 (23)	Isorhamnetin-3-*O*-glucuronide
7	16.34	329	623	461 (21), 315 (5), 179 (15), 161 (100)	Verbascoside
8	16.59	329	623	461 (18), 315 (8), 179 (11), 161 (100)	Isoverbascoside
9	18.51	329	637	461 (18), 315 (13), 193 (31), 175 (100), 161 (25)	Eukovoside

**Table 4 pharmaceuticals-17-00593-t004:** Results of extraction runs for the three matrices under study applying HAE and UAE and for the two dependent variables evaluated.

Run	Cistus				Acacia				Lemon verbena			
	HAE		UAE		HAE		UAE		HAE		UAE	
	R_1_	R_2_	R_1_	R_2_	R_1_	R_2_	R_1_	R_2_	R_1_	R_2_	R_1_	R_2_
	TPu	DR	TPu	DR	TPro	DR	TMy	DR	TMar	DR	TFor	DR
1	263.53	0.22	225.20	0.22	64.40	0.20	84.73	0.10	71.26	0.32	76.66	0.24
2	209.57	0.26	233.09	0.20	140.42	0.13	50.22	0.13	0.00	0.12	78.76	0.11
3	225.32	0.25	211.70	0.09	113.87	0.20	97.90	0.11	88.65	0.32	77.69	0.11
4	211.90	0.29	213.34	0.30	75.04	0.12	101.17	0.01	83.32	0.32	70.86	0.11
5	126.46	0.16	246.13	0.26	113.57	0.27	77.48	0.04	0.00	0.11	21.48	0.25
6	265.13	0.23	226.14	0.30	123.02	0.21	99.30	0.13	94.73	0.33	82.73	0.03
7	253.19	0.28	199.19	0.34	217.94	0.32	81.56	0.04	102.06	0.25	41.43	0.21
8	107.46	0.21	180.46	0.11	105.33	0.29	82.09	0.14	88.24	0.29	100.48	0.04
9	229.03	0.24	204.18	0.27	676.17	0.24	83.53	0.10	0.00	0.08	78.62	0.12
10	205.68	0.26	275.01	0.18	193.21	0.27	86.65	0.03	112.84	0.24	74.87	0.09
11	217.11	0.35	211.55	0.16	172.22	0.34	51.37	0.13	90.23	0.32	76.59	0.26
12	250.86	0.30	235.67	0.87	188.66	0.07	86.06	0.12	0.00	0.15	43.44	0.21
13	259.72	0.34	159.03	0.28	185.27	0.15	91.74	0.06	78.93	0.23	78.42	0.28
14	240.71	0.34	236.96	0.28	119.42	0.27	85.34	0.03	98.67	0.30	72.87	0.12
15	65.97	0.12	195.53	0.20	152.34	0.33	108.82	0.02	88.80	0.07	82.75	0.03
16	217.03	0.33	267.06	0.18	153.12	0.17	96.84	0.23	0.00	0.29	29.29	0.23
17	236.52	0.24	44.97	0.08	152.07	0.22	101.35	0.10	105.75	0.25	64.98	0.17
18	58.78	0.24	201.78	0.10	111.69	0.22	64.48	0.01	91.93	0.29	80.28	0.15
19	229.32	0.14	203.02	0.22	158.43	0.36	100.86	0.10	104.21	0.29	80.61	0.10
20	234.27	0.25	255.98	0.21	139.83	0.25	44.21	0.09	100.27	0.32	71.26	0.03
21	227.02	0.29	25.12	0.14	162.17	0.10	97.29	0.09	64.52	0.32	74.02	0.10
22	202.29	0.34	169.65	0.28	133.02	0.05	94.76	0.19	94.34	0.31	85.43	0.16
23	144.53	0.19	193.51	0.27	160.04	0.18	116.51	0.02	84.89	0.30	80.40	0.10
24	235.65	0.33	251.70	0.07	118.77	0.24	81.75	0.02	91.46	0.26	80.04	0.16
25	85.74	0.17	220.82	0.24	121.89	0.18	46.79	0.11	0.00	0.14	94.01	0.07
26	225.96	0.28	26.79	0.14	154.03	0.15	11.61	0.10	100.37	0.30	80.53	0.27
27	212.42	0.32	186.07	0.20	152.64	0.17	31.19	0.08	83.46	0.26	45.73	0.21
28	187.92	0.28	154.16	0.18	139.14	0.24	87.75	0.09	97.33	0.29	89.44	0.03
29	244.95	0.21	162.29	0.04	180.16	0.27	45.15	0.13	86.60	0.40	33.49	0.25

TPu—total punicalagin; DR—extraction yield; TPro—total procyanidin; TMy—total myricetin; TMar—total martynoside; TFor—total forsythiaside.

**Table 5 pharmaceuticals-17-00593-t005:** Coded equations of the HAE and UAE and two responses (R_1_—major phenolic compound; R_2_—extraction yield), as well as the most significative factors for each optimization (first, second, and third). All parameters under 0.001 were removed from the equations due to low relevance.

	Cistus	Equation
HAE	R_1_ = 218 − 79.94*EtOH* + 13.97*t* + 10.62*T* − 0.2257*S*/*L Ratio* + 7.05*EtOH*·*t* + 5.75*EtOH*·*T* − 14.14*EtOH*·*S*/*L Ratio* − 19.49*t*·*T* − 15.15*t*·*S*/*L Ratio* − 10.81*T*·*S*/*L Ratio* − 54.14*EtOH*^2^ − 4.33*t*^2^ − 1.80*T*^2^ + 7.25*S*/*L Ratio*^2^ 1st Time (*t*) 2nd Temperature (*T*) 3rd Ethanol (*EtOH*)	(1)
	R_2_ = 1.68 − 0.0321*EtOH* + 0.0210*t* + 0.0417*T* + 0.0179*EtOH*·*T* + 0.0238*EtOH*·*S*/*L Ratio* − 0.0965*EtOH*^2^ − 0.0389*t*^2^ − 0.0150*T*^2^ 1st Temperature (*T*) 2nd Time (*t*) 3rd Interaction ethanol + solid/liquid ratio (*EtOH·S*/*L Ratio*)	(2)
UAE	R_1_ = 198.13 − 32.95*EtOH* + 11.98*t* − 6.13*P* + 5.12*S*/*L Ratio* + 5.18*EtOH*·*t* + 29.63*EtOH*·*P* + 1.18*EtOH*·*S*/*L Ratio* + 3.24*t*·*P* − 4.06*t*·*S*/*L Ratio* − 107.20*P*·*S*/*L Ratio* + 33.40*EtOH*^2^ + 27.36*t*^2^ − 28.97*P*^2^ − 29.18*S*/*L Ratio*^2^ 1st Ethanol^2^ (*EtOH*^2^) 2nd Interaction ethanol and ultrasonic power (EtOH·P) *3*rd Time^2^(*t*^2^)	(3)
	R_2_ = 0.2695 − 0.0132*EtOH* + 0.0449*t* + 0.0517*P* + 0.0220*S*/*L Ratio* + 0.0328*EtOH*·*t* − 0.0219*EtOH*·*P* + 0.0305*EtOH*·*S*/*L Ratio* + 0.0758*t*·*P* + 0.0575*P*·*S*/*L Ratio* − 0.0655*EtOH*^2^ − 0.0114*t*^2^ − 0.0863*P*^2^ − 0.0168*S*/*L Ratio*^2^ 1st Interaction time and ultrasonic power (*t*·*P*) 2nd Interaction ultrasonic power and solid/liquid ratio (*P*·*S*/*L Ratio*) 3rd Ultrasonic power (*P*)	(4)
	**Acacia**	
HAE	R_1_ = 121.53 + 30.06*EtOH* − 2.46*t*+ 7.03*T* − 12.26*S*/*L Ratio* + 9.41*EtOH*·*t* − 13.46*EtOH*·*T* − 18.57*EtOH*·*S*/*L Ratio* + 13.48*t·T* − 32.60*t*·*S*/*L Ratio* − 59.64*T*·*S*/*L Ratio* − 11.89*EtOH*^2^ − 1.51*t*^2^ + 33.81*T*^2^ + 41.99*S*/*L Ratio*^2^ 1st Solid/liquid Ratio^2^ (*S*/*L Ratio*^2^) 2nd Interaction time and temperature (*t·T*) 3rd Interaction ethanol and time (*EtOH·t*)	(5)
	R_2_ = 0.2707 − 0.0294*EtOH* + 0.0591*t* + 0.0665*T* − 0.0167*S*/*L Ratio* + 0.0109*EtOH*·*t* + 0.0268*EtOH*·*T* + 0.0332*t*·*T* − 0.0218*t*·*S*/*L Ratio* − 0.0841*EtOH*^2^ − 0.0538*t*^2^ − 0.0118*T*^2^ 1st Temperature (*T*) 2nd Time (*t*) 3rd Interaction time and temperature (*t*·*T*)	(6)
UAE	R_1_ = 92.93 + 17.36*EtOH* − 0.7926*t* − 2.09*P* + 0.8464*S*/*L Ratio* − 4.97*EtOH*·*t* + 2.48*EtOH*·*P* + 0.9422*EtOH*·*S*/*L Ratio* + 4.72*t*·*P* + 3.27*t·S*/*L Ratio* − 4.19*P*·*S*/*L Ratio* − 32.57*EtOH*^2^ − 2.57*t*^2^ + 4.13*P*^2^ + 3.91*S*/*L Ratio*^2^ 1st Ethanol (*EtOH*) 2nd Interaction time and ultrasonic power (*t*·*P*) 3rd Interaction time and solid/liquid ratio (*t·S*/*L Ratio*)	(7)
	R_2_ = 0.1022 − 0.0261*EtOH* + 0.0148*t* + 0.0633*P* − 0.0316*S*/*L Ratio* + 0.0432*EtOH*·*P* − 0.0249*EtOH*·*S*/*L Ratio* + 0.0237*t*·*P* − 0.0134*EtOH*^2^ − 0.0263*t*^2^ + 0.0153*S*/*L Ratio*^2^ 1st Ultrasonic power (*P*) 2nd Interaction ethanol and ultrasonic power (*EtOH·P*) 3rd Interaction ethanol and solid/liquid ratio (*EtOH*·*S*/*L Ratio*)	(8)
	**Lemon verbena**	
HAE	R_1_ = 91.84 − 49.65*EtOH* + 2.08*t* − 2.86*T* − 2.03*S*/*L Ratio* + 1.18*EtOH*·*t* + 17.28*EtOH*·*T* + 3.30*EtOH*·*S*/*L Ratio* − 5.89*t*·*T* − 4.76t·*S*/*L Ratio* − 0.2039T·*S*/*L Ratio* − 44.42*EtOH*^2^ − 0.3719*t*^2^ − 4.16*T*^2^ + 0.8070*S*/*L Ratio*^2^ 1st Interaction ethanol and temperature (*EtOH*·*T*) 2nd Interaction ethanol and solid/liquid ratio (*EtOH*·*S*/*L*) 3rd Time (*t*)	(9)
	R_2_ = 0.3117 − 0.0809*EtOH* + 0.0289*t* + 0.0341*T* − 0.0161*EtOH*·*S*/*L Ratio* − 0.1126*EtOH*^2^ − 0.0148*t*^2^ 1st Temperature (*T*) 2nd Time (*t*) 3rd Interaction time and temperature (*t*·*T*)	(10)
UAE	R_1_ = 79.68 + 25.48*EtOH* + 1.82*t* − 0.0862*P* + 0.1613*S*/*L Ratio* + 6.92*EtOH*·*t* + 11.75*EtOH*·*P* − 4.71*EtOH*·*S*/*L Ratio* − 1.40*t*·*P* − 6.01*t*·*S*/*L Ratio* + 0.7934*P*·*S*/*L Ratio* − 17.52*EtOH* ^2^ + 0.5586*t*^2^ − 2.41*P*^2^ − 0.7346*S*/*L Ratio*^2^ 1st Ethanol (*EtOH*) 2nd Interaction ethanol and time (*EtOH·t*) 3rd Interaction ethanol and ultrasonic power (*EtOH*·*P*)	(11)
	R_2_ = 0.1373 − 0.0967*EtOH* + 0.0139*t* + 0.0722*P* − 0.0127*S*/*L Ratio* − 0.0113*P*·*S*/*L Ratio* + 0.0493*P*^2^ 1st Ultrasonic power (*P*) 2nd Ultrasonic power^2^ (*P*^2^) 3rd Time (*t*)	(12)

**Table 6 pharmaceuticals-17-00593-t006:** Optimized conditions of the four factors regarding each extraction type for R_1_ (major polyphenol), R_2_ (dry weight), as well as the desirability.

		HAE					UAE				
		Optimal Variable Conditions					Optimal Variable Conditions				
		*X*_1_ EtOH (%, *v*/*v*)	*X*_2_ t (min)	*X*_3_ T (°C)	*X*_4_ S/L Ratio	Predicted Quantity (mg/g dw) (g/g)	*X*_1_ EtOH (%, *v*/*v*)	*X*_2_ t(min)	*X*_3_ P(W)	*X*_4_ S/L Ratio	Predicted Quantity (mg/g dw) (g/g)
Cistus	Major phenolic compound (mg/g dw)	19	178	30	11	TPu = 266	3.22	22	171	35	TPu = 284
	Extraction yield (g/g)	48	121	80	36	DR = 0.35	55	27	421	39	DR = 0.35
	Derringer’s desirability	34	120	80	10	TPu = 262 DR = 0.35	37	30	406	25	TPu = 246 DR = 0.33
Acacia	Major phenolic compound (mg/g dw)	74	86	24	50	TPro = 233	65	8	50	50	TMy = 111
	Extraction yield (g/g)	28	160	80	12	DR = 0.37	70	20	483	10	DR = 0.24
	Derringer’s desirability	69	173	73	11	TPro = 287 DR = 0.37	65	17	500	10	TMy = 105 DR = 0.23
Lemon verbena	Major phenolic compound (mg/g dw)	13	96	49	17	TMar = 114	94	25	399	29	TFor = 101
	Extraction yield (g/g)	31	120	80	40	DR = 0.39	7	12	387	38	DR = 0.31
	Derringer’s desirability	30	120	64	10	TMar = 109 DR = 0.34	66	30	400	10	TFor = 96 DR = 0.26

TPu—total punicalagin; DR—extraction yield; TPro—total procyanidin; TMy—total myricetin; TMar—total martynoside; TFor—total forsythiaside. Desirability units are the same as R_1_ and R_2_.

**Table 7 pharmaceuticals-17-00593-t007:** 3D response chart of the heat-assisted extraction (HAE) of cistus at the optimal values.

HAE			
	R_1_—TPu	R_2_—DR	Desirability
Temperature vs. Time	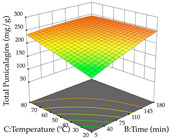	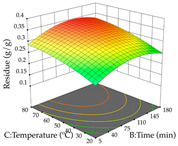	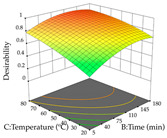
Solvent vs. Time	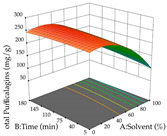	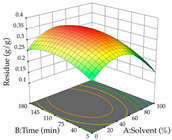	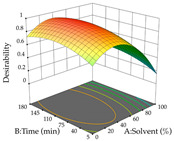
Solvent vs. Temperature	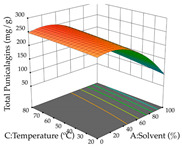	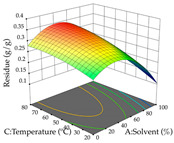	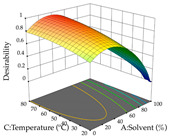
Temperature vs. Ratio	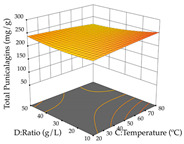	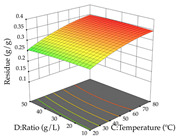	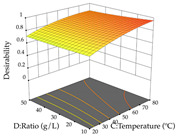
Time vs. Ratio	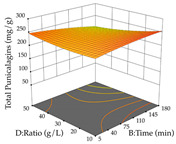	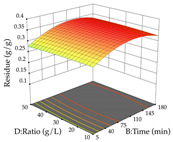	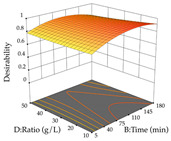
Solvent vs. Ratio	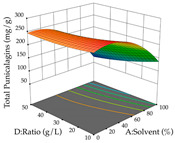	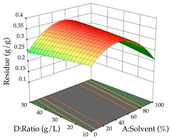	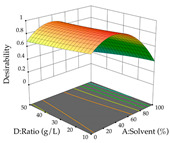

**Table 8 pharmaceuticals-17-00593-t008:** 3D response chart of the ultrasound-assisted extraction (UAE) of cistus at the optimal values.

UAE			
	R_1_—TPu	R_2_—DR	Desirability
Power vs. Time	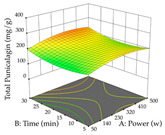	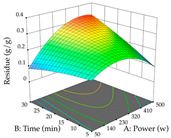	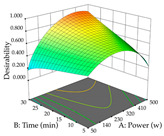
Solvent vs. Time	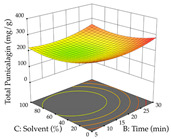	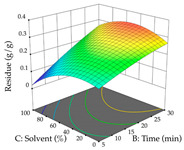	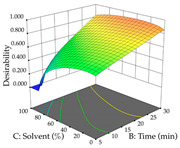
Solvent vs. Power	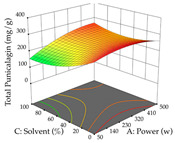	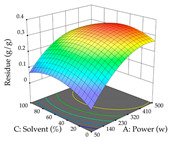	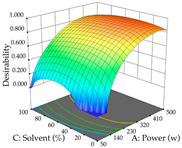
Power vs. Ratio	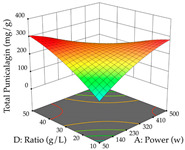	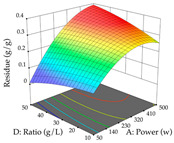	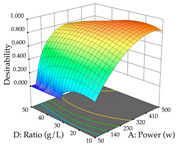
Time vs. Ratio	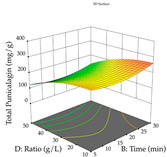	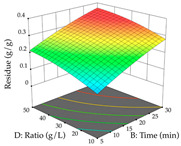	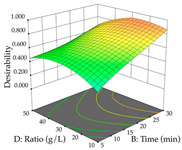
Solvent vs. Ratio	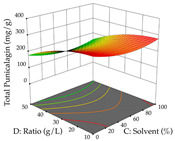	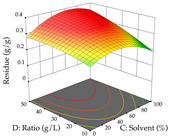	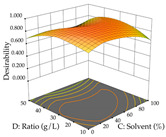

**Table 9 pharmaceuticals-17-00593-t009:** 3D response chart of the heat-assisted extraction (HAE) of acacia at the optimal values.

HAE			
	R_1_—TPro	R_2_—DR	Desirability
Temperature vs. Time	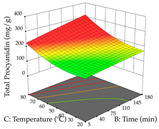	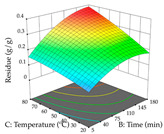	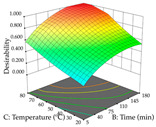
Solvent vs. Time	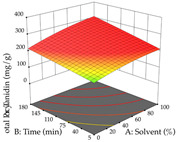	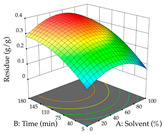	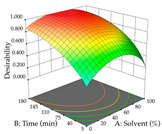
Solvent vs. Temperature	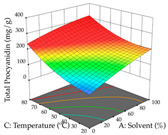	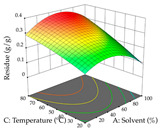	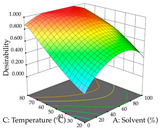
Temperature vs. Ratio	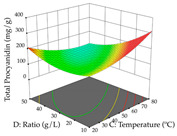	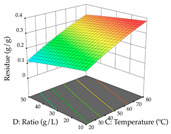	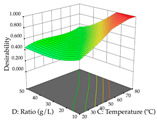
Time vs. Ratio	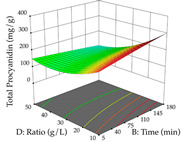	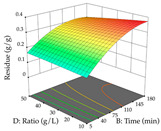	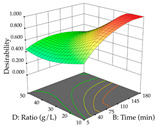
Solvent vs. Ratio	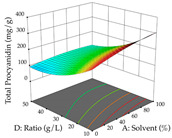	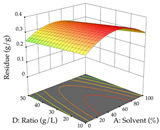	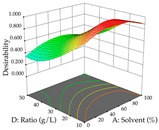

**Table 10 pharmaceuticals-17-00593-t010:** 3D response chart of the ultrasound-assisted extraction (UAE) of acacia at the optimal values.

UAE			
	R_1_—TMy	R_2_—DR	Desirability
Power vs. Time	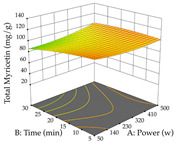	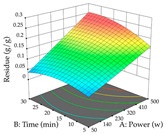	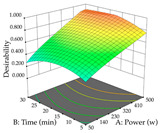
Solvent vs. Time	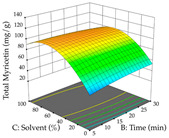	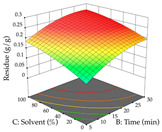	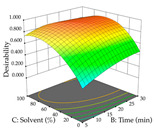
Solvent vs. Power	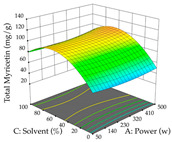	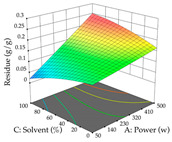	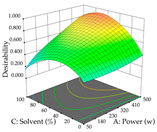
Power vs. Ratio	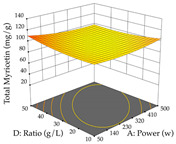	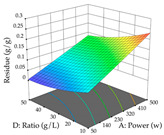	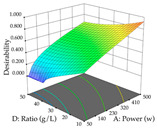
Time vs. Ratio	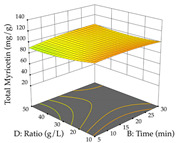	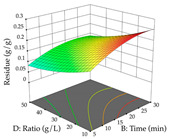	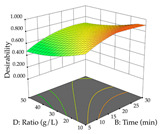
Solvent vs. Ratio	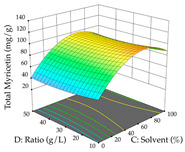	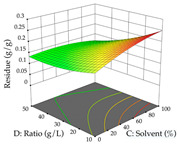	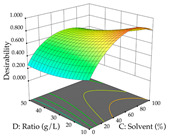

**Table 11 pharmaceuticals-17-00593-t011:** 3D response chart of the heat-assisted extraction (HAE) of lemon verbena at the optimal values.

HAE			
	R_1_—Mar	R_2_—DR	Desirability
Temperature vs. Time	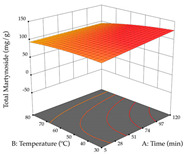	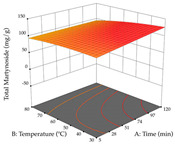	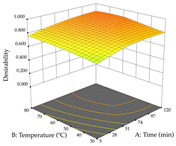
Solvent vs. Time	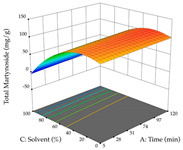	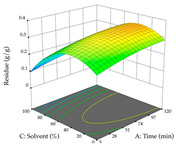	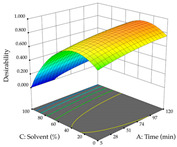
Solvent vs. Temperature	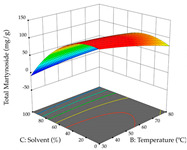	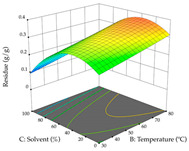	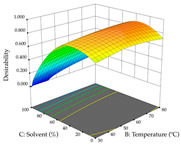
Temperature vs. Ratio	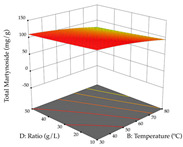	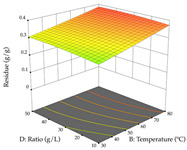	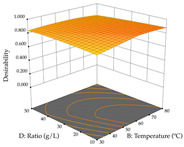
Time vs. Ratio	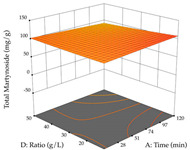	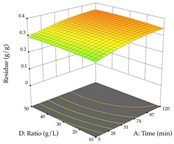	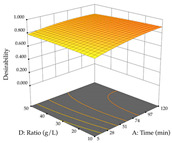
Solvent vs. Ratio	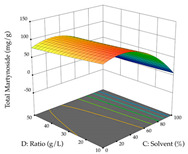	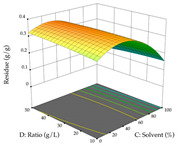	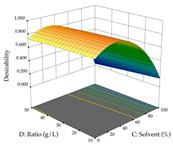

**Table 12 pharmaceuticals-17-00593-t012:** 3D response chart of the ultrasound-assisted extraction (UAE) of lemon verbena at the optimal values.

UAE			
	R_1_—TFor	R_2_—DR	Desirability
Power vs. Time	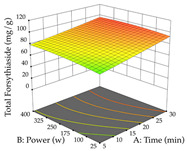	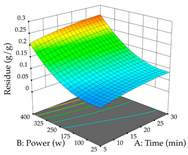	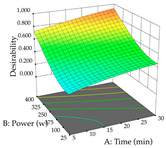
Solvent vs. Time	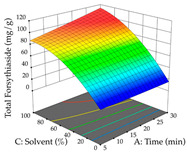	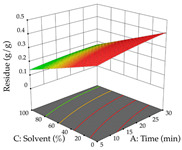	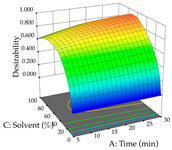
Solvent vs. Power	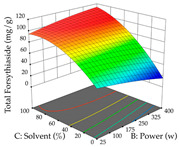	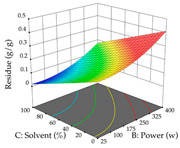	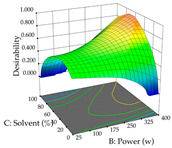
Power vs. Ratio	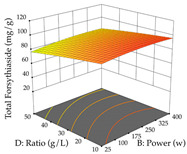	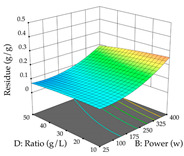	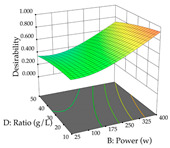
Time vs. Ratio	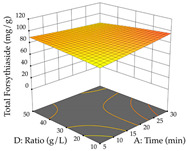	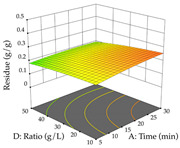	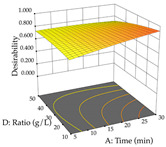
Solvent vs. Ratio	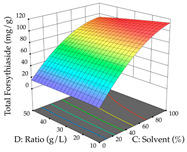	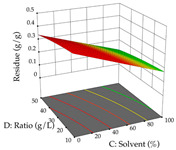	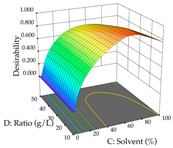

**Table 13 pharmaceuticals-17-00593-t013:** Independent variables of HAE and UAE for Box–Behnken design for cistus leaves.

	HAE				UAE			
Run	*X*_1_ EtOH (%, *v*/*v*)	*X*_2_ t (min)	*X_3_* T (°C)	*X*_4_ S/L Ratio	*X*_1_ P (W)	*X*_2_ t (min)	*X_3_* EtOH (%, *v*/*v*)	*X_4_* S/L Ratio
1	0	92.5	50	50	275	30	0	30
2	50	5	50	50	275	17.5	0	50
3	50	92.5	20	50	50	30	50	30
4	50	180	50	50	500	17.5	50	50
5	100	92.5	50	10	275	30	100	30
6	0	92.5	20	30	275	30	50	50
7	50	5	80	30	500	30	50	30
8	100	92.5	50	50	275	17.5	100	10
9	50	180	20	30	275	17.5	50	30
10	50	92.5	20	10	275	5	0	30
11	50	92.5	80	50	50	5	50	30
12	50	180	50	10	500	17.5	50	10
13	50	180	80	30	275	17.5	50	30
14	50	92.5	80	10	275	17.5	50	30
15	100	5	50	30	500	17.5	0	30
16	50	92.5	50	30	50	17.5	0	30
17	0	180	50	30	50	17.5	100	30
18	100	92.5	80	30	275	5	100	30
19	100	92.5	20	30	275	5	50	50
20	0	92.5	80	30	275	17.5	0	10
21	50	92.5	50	30	500	17.5	100	30
22	50	92.5	50	30	275	17.5	50	30
23	50	5	20	30	275	30	50	10
24	50	92.5	50	30	50	17.5	50	50
25	100	180	50	30	275	17.5	50	30
26	0	92.5	50	10	50	17.5	50	10
27	50	92.5	50	30	500	5	50	30
28	50	5	50	10	275	5	50	10
29	0	5	50	30	275	17.5	100	50

EtOH—ethanol proportion; t—time; T—temperature; S/L Ratio—solid–liquid ratio; P—ultrasonic power.

**Table 14 pharmaceuticals-17-00593-t014:** Independent variables of HAE and UAE for Box–Behnken design for acacia leaves.

	HAE				UAE			
Run	*X*_1_ EtOH (%, *v*/*v*)	*X*_2_ t (min)	*X*_3_ T (°C)	*X*_4_ S/L Ratio	*X*_1_ P (W)	*X*_2_ t (min)	*X_3_* EtOH (%, *v*/*v*)	*X*_4_ S/L Ratio
1	0	180	50	30	275	17.5	50	30
2	100	92.5	50	50	275	17.5	0	10
3	0	92.5	50	50	275	17.5	50	30
4	0	5	50	30	50	30	50	30
5	50	180	20	30	275	30	100	30
6	50	92.5	20	10	500	17.5	50	50
7	50	180	50	10	500	17.5	100	30
8	50	92.5	50	30	275	30	50	10
9	50	92.5	20	50	500	5	50	30
10	50	92.5	50	30	275	17.5	100	10
11	50	92.5	80	10	500	17.5	0	30
12	100	92.5	20	30	275	30	50	50
13	50	5	50	50	275	5	50	50
14	50	92.5	50	30	275	17.5	100	50
15	50	92.5	80	50	50	5	50	30
16	50	5	80	30	500	17.5	50	10
17	100	92.5	80	30	275	17.5	50	30
18	0	92.5	50	10	50	17.5	100	30
19	50	180	80	30	275	5	50	10
20	50	92.5	50	30	50	17.5	0	30
21	50	5	20	30	50	17.5	50	10
22	100	5	50	30	500	30	50	30
23	100	180	50	30	50	17.5	50	50
24	50	180	50	50	275	5	100	30
25	0	92.5	20	30	275	30	0	30
26	50	5	50	10	275	17.5	50	30
27	100	92.5	50	10	275	5	0	30
28	0	92.5	80	30	275	17.5	50	30
29	50	92.5	50	30	275	17.5	0	50

EtOH—ethanol proportion; t—time; T—temperature; S/L Ratio—solid–liquid ratio; P—ultrasonic power.

**Table 15 pharmaceuticals-17-00593-t015:** Independent variables of HAE and UAE for Box–Behnken design for lemon verbena.

	HAE				UAE			
Run	*X*_1_ EtOH (%, *v*/*v*)	*X*_2_ t (min)	*X*_3_ T (°C)	*X*_4_ S/L Ratio	*X*_1_ P (W)	*X*_2_ t (min)	*X_3_* EtOH (%, *v*/*v*)	*X*_4_ S/L Ratio
1	50	62.5	55	30	400	5	50	30
2	100	120	55	30	25	17.5	50	10
3	50	62.5	55	30	25	17.5	50	50
4	50	120	55	50	212.5	5	50	10
5	100	62.5	55	50	400	17.5	0	30
6	50	62.5	80	10	212.5	5	100	30
7	0	5	55	30	212.5	17.5	0	50
8	50	62.5	30	50	212.5	30	100	30
9	100	5	55	30	212.5	17.5	50	30
10	0	62.5	30	30	25	5	50	30
11	50	62.5	80	50	400	30	50	30
12	100	62.5	80	30	212.5	5	0	30
13	50	5	30	30	400	17.5	50	10
14	50	62.5	55	30	212.5	30	50	50
15	100	62.5	30	30	212.5	17.5	100	50
16	0	62.5	55	50	212.5	17.5	0	10
17	0	62.5	55	10	212.5	30	50	10
18	50	62.5	30	10	212.5	17.5	50	30
19	50	120	30	30	212.5	5	50	50
20	50	62.5	55	30	25	17.5	100	30
21	0	62.5	80	30	212.5	17.5	50	30
22	50	120	55	10	212.5	17.5	50	30
23	50	5	80	30	25	30	50	30
24	50	5	55	50	212.5	17.5	50	30
25	100	62.5	55	10	400	17.5	100	30
26	50	62.5	55	30	400	17.5	50	50
27	50	5	55	10	25	17.5	0	30
28	0	120	55	30	212.5	17.5	100	10
29	50	120	80	30	212.5	30	0	30

EtOH—ethanol proportion; t—time; T—temperature; S/L Ratio—solid–liquid ratio; P—ultrasonic power.

## Data Availability

The original contributions presented in the study are included in the article further inquiries can be directed to the corresponding author.

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
