# Peer review of "Cistus, Acacia, and Lemon verbena Valorization through Response Surface Methodology: Optimization Studies and Potential Application in the Pharmaceutical and Nutraceutical Industries"

_pharmaceuticals, 2024, doi:10.3390/ph17050593_

Round 1

Reviewer 1 Report

Comments and Suggestions for Authors

The manuscript from Fernandes et al. describes the process of optimizing the extraction of bioactive substances from cistus, acacia and lemon verbena leaves.

In general, the article is well written and provides a lot of valuable information regarding the chemical composition of the above-mentioned raw materials. Another strength of the manuscript is the well-developed and described chromatographic analysis.

However, I have a few issues that should be improved:

1.       Table 6. Variable X3 (ultrasound treatment) should include ultrasonic intensity, not temperature.

2.       In table 5 it would make more sense to provide equations based on the uncoded values of the variables (real)

3.       There are generally no tables with statistical processing. It is important which variables are statistically significant and which are not. The models in Table 5 should include only main effects and interactions that are statistically significant.

4.       The authors determined the optimal extraction conditions in the tested process. However, there is no verification of the model based on experimental data.

5.       Methodology. Please indicate what the ultrasound intensity was during the extraction process.

6.       The conclusions should be reworked. Now this is a summary of the article rather than conclusions.

Author Response

The manuscript from Fernandes et al. describes the process of optimizing the extraction of bioactive substances from cistus, acacia and lemon verbena leaves.

In general, the article is well written and provides a lot of valuable information regarding the chemical composition of the above-mentioned raw materials. Another strength of the manuscript is the well-developed and described chromatographic analysis.

However, I have a few issues that should be improved:

  1. Table 6. Variable X3 (ultrasound treatment) should include ultrasonic intensity, not temperature.

Answer: The unit was changed to ultrasonic intensity.

  1. In table 5 it would make more sense to provide equations based on the uncoded values of the variables (real)
  2. There are generally no tables with statistical processing. It is important which variables are statistically significant and which are not. The models in Table 5 should include only main effects and interactions that are statistically significant.
  3. The authors determined the optimal extraction conditions in the tested process. However, there is no verification of the model based on experimental data.
  4. Please indicate what the ultrasound intensity was during the extraction process.
  5. The conclusions should be reworked. Now this is a summary of the article rather than conclusions.

In table 5 it would make more sense to provide equations based on the uncoded values of the variables (real)

Answer: The table was changed to include the uncoded values.

There are generally no tables with statistical processing. It is important which variables are statistically significant and which are not. The models in Table 5 should include only main effects and interactions that are statistically significant.

Answer: While the authors totally agree with the reviewers point, they believe that all factors should be in the table. Thus, as a compromise, they added the three most significant factors for each optimization in table 5, below the equations.

The authors determined the optimal extraction conditions in the tested process. However, there is no verification of the model based on experimental data.

Answer: The extraction conditions were tested preliminary, but a thorough research on their results is still ongoing, and a full publication on their testing and results is being prepared, hence the lack of verification in this publication.

Methodology. Please indicate what the ultrasound intensity was during the extraction process.

Answer: A specific intensity cannot be added, as the intensity in Watts was one parameter that varied in the optimization, ranging from 50 to 500W.

The conclusions should be reworked. Now this is a summary of the article rather than conclusions.

Answer: The conclusions were improved based on the reviewer’s request.

Reviewer 2 Report

Comments and Suggestions for Authors

Dear Authors,

The selected plant species for extraction procedures are interesting and underused which qualify them for proposed valorization. However, it is extremely difficult to make the text (especially Introduction and Results) cohesive when chosen species are not from the same family nor have any other common characteristic. Further, it makes very difficult for a reader to comprehend why you combined those species in one paper. I would suggest complete reconstruction of the Introduction with attempt to find some rationale for study in question.

The detailed comments and suggestions are attached in the docx file below.

Kind regards

Comments on the Quality of English Language

Moderate editing of English language required.

Author Response

Dear Authors,

The selected plant species for extraction procedures are interesting and underused which qualify them for proposed valorization. However, it is extremely difficult to make the text (especially Introduction and Results) cohesive when chosen species are not from the same family nor have any other common characteristic. Further, it makes very difficult for a reader to comprehend why you combined those species in one paper. I would suggest complete reconstruction of the Introduction with attempt to find some rationale for study in question.

The detailed comments and suggestions are attached in the docx file below.

Kind regards

Introduction: complete reconstruction of the Introduction with attempt to find some rationale for study in question.

Answer: As suggested, a justification for the study of the three plants in question was added to the introduction (line 45-51).

Specific remarks

Line 26, 232, 262, 278, 329, 664 – extraction yield

Answer: The change was performed.

Line 30 - HAE, permutation

Answer: The change was performed.

Line 39 - The sentence probably should go: “they were used not only as food, but…”.

Answer: The change was performed

Line 60 – Verbenaceae

Answer: The change was performed

Line 45 - 66 – Please try to find some common characteristics between these plant species, except that they are underused, in order to make introduction more consistent and meaningful.

Answer: The change was performed, line 45-51.

Line 82 The title of section 2.1. is the same as 2.2.

Answer: The change of title of section 2.2 (line 223) was performed.

Line 125 - 131 – Too long sentence.

Answer: The change was performed.

Line 179 – Unnecessary sentence.

Answer: The sentence was removed.

Line 184 – Was

Answer: The change was performed.

Line 212 - 216 – Used; Supercritical fluid extraction promotes extraction of lipophilic compounds primarily; Split this sentence in two.

Answer: The changes were performed.

Line 240, 257, 267, 286 – redundant

Answer: The sentence was changed.

Line 244 – compound

Answer: The change was performed.

Line 247, 261 – HAE and UAE

Answer: The change was performed.

Line 249 - 250 – This should be put prior to Table 4.

Answer: The sentence was changed to line 236 and 237.

Table 5 – You can exclude parameters and their interactions which are not relevant

Answer: As requested by another reviewer, the authors reached a compromise to leave all factors in the equations and point out the three most significant factors below the equations in table 5.

Line 266 – Both extraction types.

Answer: The change was performed.

Line 266 - 269 – Out of this part one could conclude that you had 12 extractions in total, while in the material section it is stated that you have performed 29 extractions. Which is it?

Answer: The sentence was changed to improve clarity, as it could cause confusion. In fact, the 29 extraction were needed to reach the results presented in the manuscript.

Line 273 – “that for obtaining”

Answer: The change was performed.

Line 274 – These are three factors, not one, which one stands out?

Answer: What stood out was ethanol. The change was performed.

Line 275 – Which two factors?

Answer: The factors were ethanol and temperature, which were included in the sentence.

Line 264 - 283 – This section (264-283) needs to improve on clarity and the flow of the information.

Answer: The section was rephrased to become more intelligible.

Line 290 – what are global responses?

Answer: Global response was a synonym for desirability, but to reduce confusion it was removed from the manuscript.

Line 292 – This data is insufficient as it was not stated out of what volume was this mass obtained.

Answer: The units are stated in table 6, namely mg/g for the major phenolic compounds and g/g for dry residue.

Line 293, 294, 299, 302, 328 – min

Answer: The change was performed.

Line 304 and 305 – This should be written in Methods.

Answer: The change was performed. Information added in section 1.6, line 661.

Line 310 – Derringer׳s desirability function allows the analyst to find the experimental conditions (factor levels) to reach, simultaneously, the optimal value for all the evaluated variables, including the researcher׳s priorities during the optimization procedure, D

Answer: The sentence was changed according to the reviewer’s suggestion, lines 31-319.

Line 316 and 317 – exclude from Tables title

Answer: Deleted.

Table 6 – Predicted Quantity - Unit?; Look comment f38

Answer: “Predicted quantity” is the predicted amount of major phenolics and dry residue that is expected to be extracted using the optimal conditions the model reached. Considering desirability, Derringer was added.

Line 322 – extraction parameters

Answer: The change was performed.

Line 478 – This section should be merged in the Introduction, after description of the plant species studied

Answer: The sentences were written in line 74 – 80.

Line 487 – instead emergent put green

Answer: The change was performed.

Line 595 – This is too general, please specify chemicals and standards that were used.

Answer: The information has been added.

Line 604 – S/L ratio

Answer: The change was performed.

Line 614 – The

Answer: The change was performed.

Line 664 – Usually, independent variables are combined with monitored responses in the Table

Answer: The authors agree with the reviewer, but in this case, as there are many columns and numbers for variables and responses the authors prefer to separate the information in two tables.

Table 13, 14 and 15 – independent variables of HAE and UAE for Box-Behnken design for citrus leaves

Answer: The change was performed.

Line 694, 702, 707 and 710 – The Conclusion is too general and too extensive without clear points; redundant in conclusion; medium polar compounds

Answer: The conclusion was restructured.

Line 700 – In the Methods it is stated that two wild species were used?

Answer: The information was updated.

Reduce the self-citation rate.

We noticed that there are self-citations in this paper.

There are 8 self-citations out of 47 total references. It is 17.02%.

Usually, the self-citation rate should be below 17.02%. Could you please revise it accordingly during the revision stage? 

Answer: Self-citation was reduced.

Round 2

Reviewer 2 Report

Comments and Suggestions for Authors

Dear Authors,

There is a noticable improvement of the quality of proposed paper after adoption of suggestions. There are a few suggestions that you should consider, one of them is to simplify rsm equations by excluding non-relevant parameters. Please find other comments in the file below.

Kind regards

Comments on the Quality of English Language

Minor editing of English language required.

Author Response

Dicotyledonous plants?

R: Yes, Magnoliopsida are dicotyledons. This information was added to the manuscript.

This Table5. Needs to be simplified, exclude non-relevant parameters.

R: All parameters below 0.001 were removed from the equations, as recommended.

Unit needs to be corrected.

R: The units were corrected.

First time mentioned in conclusion.

R: Rockrose was removed for clarity and swapped for Cistus.